# HETEGRAPH-MAMBA:
# HETEROGENEOUS GRAPH LEARNING VIA SELECTIVE STATE SPACE MODEL

## ABSTRACT

We propose a heterogeneous graph mamba network (HGMN) as the first exploration in leveraging the selective state space models (SSSMs) for heterogeneous graph learning. Compared with the literature, our HGMN overcomes two major challenges: (i) capturing long-range dependencies among heterogeneous nodes and (ii) adapting SSSMs to heterogeneous graph data. Our key contribution is a general graph architecture that can solve heterogeneous nodes in real-world scenarios, followed an efficient flow. Methodologically, we introduce a two-level efficient tokenization approach that first captures long-range dependencies within identical node types, and subsequently across all node types. Empirically, we conduct comparisons between our framework and 19 state-of-the-art methods on the heterogeneous benchmarks. The extensive comparisons demonstrate that our framework outperforms other methods in both the accuracy and efficiency dimensions.

## 1 INTRODUCTION

This work proposes a heterogeneous graph mamba network (HGMN) to explore the next-generation heterogeneous graph learning by leveraging the powerful selective state space models (SSSMs), tailored to surpass the most popular transformer-based methods. This work addresses two major challenges in devising SSSMs-enhanced solution that needs greater expressiveness and minimized inference time for heterogeneous graph tasks: (i) **long-range dependencies**, for example, IMDB's sparse network of 21K nodes with only 87K edges requires leveraging distant neighbor information to enhance node embeddings— a challenge amplified by the heterogeneity of the graph and (ii) **graph-to-sequence conversion**, where the process maps unordered graph data into a sequential structure, leveraging heterogeneous graph characteristics for effective SSSM processing.

To capture the long-range dependencies among graph, several studies (Yun et al., 2020; Hu et al., 2020b) integrate the transformer architecture, which utilizes full global attention to enhance the representation of diverse node and edge types. HINormer(Mao et al., 2023) further extends this approach by proposing a global-range attention mechanism coupled with a dual-encoder to manage heterogeneous and structural data within graph representations. However, models employing global attention mechanisms always require a quadratic time complexity of $O(n^2)$, which limits their scalability. To address it, models like Exphormer (Shirzad et al., 2023) employ sparse attention techniques, such as random subsampling, to reduce computational demands, though they still face significant challenges in robustness. Recently, Mamba (Gu & Dao, 2023), an enhanced state space model(SSM), introduces a data-dependent state transition mechanism that not only captures long-range context but also demonstrates linear-time efficiency and competitive performance with traditional Transformers. This innovation inspires researchers to adapt its architecture on many domains (Jiang et al., 2024; Liu et al., 2024; Zhang et al., 2024) and get surprising performance. Additionally, some studies (Wang et al., 2024; Behrouz & Hashemi, 2024) propose effective strategies for mapping graph structures into ordered sequences for integrating Mamba block. Despite these achievements, the complexity of real-world heterogeneous graph scenarios remains a challenge, prompting us to propose a novel graph-to-sequence conversion mechanism to better capture long-range heterogeneity dependencies among such graphs.

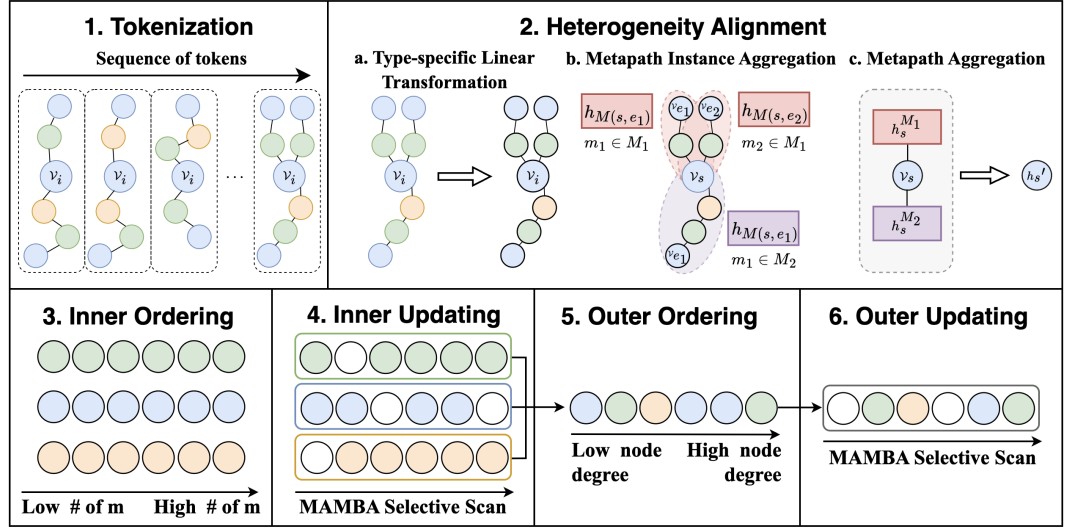

Figure 1: **Overview**. (1) Input graph is sequentialized into tokens, each being a graph of metapath instances centered on a target node $\mathcal{V}_i$. (2a) Nodes of node type $A \in \mathcal{A}$ are projected onto the same latent representation space with a type-specific linear transformation. (2b) Node representations within each metapath instance $m$, which is a node sequence $M(s, e)$ from start node $\mathcal{V}_s$ to end node $\mathcal{V}_e$, are aggregated including the intermediate nodes. An instance encoder is used to produce vector representations $h_{M(s,e)}$ for each $m$, embedding in-between context as well. For every $\mathcal{V}_s$, the significance of each $m \in M$ is modeled through a graph attention layer to form vector representation $h_s^{M_i}$. (2c) All $h_s^{M_i}$ are aggregated to capture varying contributions of each $M \in \mathcal{M}$. (3) Nodes grouped by node type are ordered in increasing numbers of $m$. (4) Context-aware filtering is applied to the groups of nodes with a MAMBA layer for each. (5) Nodes are re-ordered in increasing order of m across all $A$. (6) Finally, a single MAMBA layer is applied to all nodes.

To address the above challenges, we introduce **HGMN**, a heterogeneous graph mamba network that provides a data-dependent alternative for attention sparsification, capable of capturing long-range heterogeneity dependencies and reducing computational costs in large heterogeneous graphs. As depicted in Figure 1, **HGMN** features a six-step scalable, flexible, and powerful architecture:

1. *Tokenization*: Map a graph into a sequence of tokens, adapting sequential encoders for graphs, where each token is a subgraph of a target node and its meta-path instances.

2. *Heterogeneity Alignment*: Project different node types into a unified latent representation space, and each token (subgraph) is aggregated into an updated target node representation.

3. *Inner Ordering*: Group nodes by node type and order within each group by the number of meta-path instances, reflecting the importance of the nodes in the same node type.

4. *Inner Updating*: Scans and selects relevant nodes for updating in each group of nodes to get the long-range homogeneity dependencies by a Mamba layer.

5. *Outer Ordering*: Order all types of nodes together by node degree, reflecting the global importance of nodes in a graph.

6. *Outer Updating*: Apply the updating process across all node types to ensure comprehensive updates and capture the long-range heterogeneity dependencies by a Mamba layer.

In the end, **HGMN** produces updated node embeddings for downstream tasks such as node prediction.

In summary, **HGMN** is the first model to capture long-range heterogeneity dependencies among large heterogeneous graphs with the support of enhanced data-dependent SSMs, delivering linear-time efficiency and satisfying performance. Our main contributions are as follows:

- We introduce **HGMN**, a novel architecture that integrates the powerful capabilities of SSSMs for addressing the unique challenges of heterogeneous graph data, particularly in improving expressiveness and minimizing inference time.

- We propose a unique **graph-to-sequence conversion mechanism** that transforms unordered graph data into a sequential format, leveraging the inherent characteristics of heterogeneous graphs to facilitate effective SSSM processing.

- We design a **six-step scalable recipe** for HGMN, which includes Tokenization, Heterogeneity Alignment, Inner and Outer Ordering, and Inner and Outer Updating steps to manage and update node representations, capturing both homogeneity and heterogeneity dependencies.

- Experimental results demonstrate that our framework can **outperform existing transformer-based and sparse attention methods** on public benchmarks, achieving higher accuracy and efficiency in public benchmarks including normal and large heterogeneous graphs.

## 2 RELATED WORK

### 2.1 HOMOGENEOUS GRAPH NEURAL NETWORKS

Homogeneous graph neural networks propose a massage-passing mechanism that aims to process the unstructured data (graph). GCN(Kipf & Welling, 2017) applies convolution directly on graphs to efficiently capture node interrelations for node classification and link prediction. GAT(Veličković et al., 2018) adds attention mechanisms to dynamically prioritize and aggregate neighboring node features in graph networks. To deal with large-scale graphs, GraphSAGE(Hamilton et al., 2017) first uses neighborhood sampling and aggregation. While these methods can get satisfying performance for some public benchmarks, the real world has more complex heterogeneous graph structures than homogeneous ones.

### 2.2 HETEROGENEOUS GRAPH NEURAL NETWORKS

To solve heterogeneous edges, RGCN(Schlichtkrull et al., 2018) adapts GCNs for multi-relational graphs by assigning distinct weights to different relationship types. RSHN(Zhu et al., 2019) embeds both nodes and edges by integrating a Coarsened Line Graph with a HGNN(Zhang et al., 2019), capturing deep relational structures in large-scale networks. HetSANN(Hong et al., 2019) directly encodes HGNNs' structural information using an attention mechanism, bypassing the need for metapath based preprocessing. NARS(Yu et al., 2020) enhances feature smoothing on heterogeneous graphs (HGs) by averaging features over relation-specific subgraphs for scalable and memory-efficient graph learning. MAGNN(Fu et al., 2020) refines HG embedding by aggregating complex relational paths and node content. HetGNN(Zhang et al., 2019) and HAN(Wang et al., 2021) apply hierarchical attention to enhance node embedding by utilizing metapath based neighbors. DiffMG(Ding et al., 2021) and PMMM(Li et al., 2022) optimize the HGNN architectures by searching for flexible and stable meta-multigraphs that capture complex semantic relations. SeHGNN(Yang et al., 2023) simplifies the HGNN architectures through a single-layer, metapath extended structure. However, addressing long-range dependencies in sparse HG like IMDB (21K nodes with only 87K edges), which demands leveraging distant neighbor information efficiently amidst heterogeneity, remains a significant challenge.

### 2.3 GRAPH TRANSFORMER

To solve the limitations of Message-Passing-based model such as expressiveness bounds, over-smoothing, over-squashing and so on, some recent work employ full global attention to enhance performance, motivated by Transformer(Vaswani et al., 2023). GTN(Yun et al., 2020) generates and utilizes new graph structures for dynamic node representation learning in heterogeneous environments without pre-defined metapaths. HGT(Hu et al., 2020b) leverages a transformer architecture to manage and enhance the representation of diverse node and edge types in heterogeneous graphs. HINormer(Mao et al., 2023) utilizes a global-range attention mechanism and dual-encoder modules to effectively manage heterogeneous and structural data in network representations. However, to achieve high expressive power, models employing global attention mechanisms like the Transformer require a time complexity of $O(n^2)$. Even though sparse attention techniques can reduce this complexity, methods like Exphormer (Shirzad et al., 2023) that use random node subsampling still face significant robustness challenges. Therefore, introducing methods capable of naturally informed context selection might present a more viable solution.

## 2.4 SELECTIVE STATE SPACE MODEL

State space models (SSMs) (Hamilton, 1994; Gu et al., 2021) recently emerge as a popular alternative to attention-based modeling architectures due to their efficiency, which stems from performing recurrent updates across sequences via hidden states. However, their effectiveness is often limited compared to Transformers because of their time-invariant transition mechanisms. To address this, Mamba (Gu & Dao, 2023) introduces a data-dependent state transition mechanism that captures long-range context, demonstrating linear-time efficiency and competitive performance with Transformers. This innovation motivates researchers to adapts its architecture across many domains (Jiang et al., 2024; Liu et al., 2024; Zhang et al., 2024). In graph domain, some studies (Wang et al., 2024; Behrouz & Hashemi, 2024) propose effective strategies for mapping graph structures into ordered sequences. Others (Li et al., 2024b; Liang et al., 2024) develop frameworks for spacial-temporal graph and point cloud tasks, showcasing the efficacy of Mamba-based models in specific benchmarks. However, the real world presents a more complex array of heterogeneous graph scenarios. It motivates us to propose a graph-to-sequence conversion mechanism to capture long-range heterogeneity dependencies among heterogeneous graphs.

## 2.5 PRELIMINARIES

**State Space Models.** The SSM-based models are inspired by the continuous system, which maps a sequence $x(t) \in \mathbb{R} \mapsto y(t) \in \mathbb{R}$ through a hidden state $h(t) \in \mathbb{R}^{\mathbb{N}}$. This system uses $\mathbf{A} \in \mathbb{R}^{\mathbb{N} \times \mathbb{N}}$ as the updating parameter and $\mathbf{B} \in \mathbb{R}^{\mathbb{N} \times 1}$, $\mathbf{C} \in \mathbb{R}^{1 \times \mathbb{N}}$ as the projection parameters.

$$h'(t) = \mathbf{A}h(t) + \mathbf{B}x(t), y(t) = \mathbf{C}h(t). \tag{1}$$

For example, S4 (Gu et al., 2021) and Mamba (Gu & Dao, 2023) models are discrete versions of the system, incorporating a timescale parameter $\mathbf{\Delta}$ that converts the continuous coefficients $\mathbf{A}$ and $\mathbf{B}$ into discrete equivalents $\overline{\mathbf{A}}$ and $\overline{\mathbf{B}}$. The transformation employs the zero-order hold (ZOH) technique, defined by the equations:

$$\overline{\mathbf{A}} = \exp\left(\mathbf{\Delta}\mathbf{A}\right), \overline{\mathbf{B}} = (\mathbf{\Delta}\mathbf{A})^{-1}(\exp\left(\mathbf{\Delta}\mathbf{A}\right) - \mathbf{I}) \cdot \mathbf{\Delta}\mathbf{B}. \tag{2}$$

Following this discretization process, the discrete form of Eq. equation 1 with a step interval of $\mathbf{\Delta}$ is as:

$$h_t = \overline{\mathbf{A}}h_{t-1} + \overline{\mathbf{B}}x_t, y_t = \mathbf{C}h_t. \tag{3}$$

Finally, the models produce the output by executing a global convolution:

$$\overline{\mathbf{K}} = (\mathbf{C}\overline{\mathbf{B}}, \mathbf{C}\overline{\mathbf{A}}\overline{\mathbf{B}}, \dots, \mathbf{C}\overline{\mathbf{A}}^{\mathbb{M}-1}\overline{\mathbf{B}}), \mathbf{y} = \mathbf{x} * \overline{\mathbf{K}}, \tag{4}$$

where $\mathbb{M}$ denotes the sequence length $\mathbf{x}$, and $\overline{\mathbf{K}} \in \mathbb{R}^{\mathbb{M}}$ represents a structured convolutional kernel.

**Definition 2.1. Heterogeneous Graph.** We define a heterogeneous graph as $\mathcal{G} = (\mathcal{V}, \mathcal{E})$, where $\mathcal{V}$ and $\mathcal{E}$ denote the sets of nodes and edges, respectively. Each node and edge is associated with predefined types through mapping functions: $\mathcal{V} \to \mathcal{A}$ for nodes and $\mathcal{E} \to \mathcal{R}$ for edges.

**Definition 2.2. Metapath.** We define a metapath $M$ of its set $\mathcal{M}$ as $A_1 \xrightarrow{R_1} A_2 \xrightarrow{R_2} \cdots \xrightarrow{R_i} A_{i+1}$ including a composite relation $R = R_1 \triangleright R_2 \triangleright \cdots \triangleright R_i$ that covers the relationships between the starting node type $A_1$ and the ending node type $A_{i+1}$, where the $\triangleright$ symbol denotes the composition.

**Definition 2.3. Metapath Instance.** We define a metapath instance $m$ of given metapath $M$ as a node sequence in the graph following the schema defined by $M$.

**Definition 2.4. Metapath Graph.** We define a metapath graph $\mathcal{G}^M$ as a subgraph constructed by nodes from all metapath-M-based node sequences in graph $\mathcal{G}$.

## 3 METHODOLOGY

### 3.1 OVERVIEW

In the ***Tokenization*** step, we map the graph into a sequence of tokens, adapting sequential encoders for graph data. Each token represents a subgraph that includes a target node and its all metapath instances,

where a metapath instance is an ordered sequence of nodes within a composite relation among the node types involved. Subsequently, in the ***Heterogeneity Alignment*** step, we use a mapping function to project nodes of different types into the same latent representation space, aggregating each token (subgraph) into an updated target node representation. Following this, in the ***Inner Ordering*** step, nodes are grouped by node type and ordered within each group based on the number of the metapath instances. This ordering reflects an assumption that the greater the number of metapath instances associated with a node, the more significant that node is within its type. In the ***Inner Update*** step, a Mamba mechanism is employed to scan and select relevant nodes for updating. Due to the recurrent updates, each token accesses information only from preceding tokens. Consequently, more important nodes are positioned closer to the end of the sequence to maximize their visibility and impact. Upon completing the updates within each node type, in the ***Outer Update*** step, all nodes are ordered by their degree, and a similar updating process is applied across all node types to ensure global updates. Ultimately, HGMN outputs the final node embeddings, which are ready for downstream tasks such as node prediction, ensuring that the system captures the long-range heterogeneity dependencies.

### 3.2 TOKENIZATION

To adapt the sequential SSM-based model for graph data, we need a process to divide a given graph into many tokens. Recent works try to employ node, edge, or even subgraph tokenization methods, each with its own advantages and disadvantages (Shirzad et al., 2023; Kim et al., 2022; He et al., 2023). In this step, we design a simple but effective and efficient strategy that considers both the heterogeneity and homogeneity of graphs.

Given a heterogeneous graph $\mathcal{G} = (\mathcal{V}, \mathcal{E})$, with nodes and edges mapped to predefined types through the functions $\mathcal{V} \rightarrow \mathcal{A}$ and $\mathcal{E} \rightarrow \mathcal{R}$, we generate a subgraph as a token for each node $\mathcal{V}_i$:

$$\mathcal{G}[\mathcal{V}_i] = \bigcup_{k=1}^{|\mathcal{M}|} \mathcal{G}[\mathcal{V}_i]^{M_k} \tag{5}$$

Each subgraph $\mathcal{G}[\mathcal{V}_i]$ includes a target node $\mathcal{V}_i$ and its metapath instances from all metapaths in the set $\mathcal{M}$. After tokenization, instead of using individual nodes, we utilize a subgraph $\mathcal{G}[\mathcal{V}_i]$ as the representative token for each node.

### 3.3 HETEROGENEITY ALIGNMENT

To aggregate diverse information, this step aligns the heterogeneity within each token (subgraph).

**Type-specific Linear Transformation.** We first address the diversity of node types by using a type-specific linear transformation to project feature vectors of different node types into the same latent representation space. This is formalized as follows for each node type $A \in \mathcal{A}$:

$$\mathbf{h}_i^A = \mathbf{W}_A \mathbf{x}_i^A + \mathbf{b}_A, \tag{6}$$

where $\mathbf{x}_i^A \in \mathbb{R}^{d_A}$ is the feature vector of node $\mathcal{V}_i$ of type $A$, and $\mathbf{W}_A \in \mathbb{R}^{d' \times d_A}$ and $\mathbf{b}_A \in \mathbb{R}^{d'}$ are the parameters of the linear transformation specific to type $A$.

**Metapath Instance Aggregation.** Following the projection, we then start to aggregate each token (subgraph) by two levels: (1) metapath instance level and (2) metapath level.

Within a metapath instance $m$ of metapath $M$, there is node sequence $M(s, e)$ including a start node $\mathcal{V}_s$, an end node $\mathcal{V}_e$, and the intermediate nodes. We use an instance encoder to transfer all the node features within $M(s, e)$ to a single vector: $\mathbf{h}_{M(s,e)} = f_\theta\left(M(s,e)\right) = f_\theta\left(\{\mathbf{h}'_t, \forall t \in \{M(s,e)\}\}\right)$ where $\mathbf{h}_{M(s,e)} \in \mathbb{R}^{d'}$ has a dimension of $d'$.

After encoding metapath instances, we utilize the graph attention layer to obtain the weighted sum of all metapath instances for each metapath $M$ related to targe node $\mathcal{V}_s$. The primary rationale is that each instance contributes to the representation of $\mathcal{V}_s$ to varying degrees. To model the significance of each metapath instance through importance weights $\alpha_{s,e}^M$, the graph attention layer firstly computes $e_{se}^M$ (quantifying the relevance of the metapath instance $M(s,e)$ to node $\mathcal{V}_s$) by $e_{se}^M = \text{LeakyReLU}\left(\mathbf{a}_M \cdot \left[\mathbf{h}'_s \| \mathbf{h}_{M(s,e)}\right]\right)$, where the attention vector $\mathbf{a}_M$, specific to metapath $M$

and parameterized in $\mathbb{R}^{2d'}$, facilitates the use of the vector concatenation operator, $\|$. Then, it models the significance of each metapath instance through importance weights $\alpha_{s,e}^M$:

$$\alpha_{se}^M = \frac{\exp\left(e_{se}^M\right)}{\sum_{e \in \mathcal{N}_s^M} \exp\left(e_{se}^M\right)}. \tag{7}$$

This significance is normalized across all possible $\mathcal{V}_e$ in $\mathcal{N}s^M$ using the softmax function. Following normalization, the importance weights $\alpha_{s,e}^M$ for each $e$ in $\mathcal{N}_s^M$ are used to form a weighted sum of the representations from the metapath instances associated with node $\mathcal{V}_s$). Finally, we aggregate these contributions by summing all instances with an activation function $\sigma(\cdot)$:

$$\mathbf{h}_s^M = \sigma\left(\sum_{e \in \mathcal{N}_s^M} \alpha_{se}^M \cdot \mathbf{h}_{M(s,e)}\right). \tag{8}$$

**Metapath Aggregation.** After aggregating the representations from all instances within each metapath, we proceed to aggregate the outputs across different metapaths for the target node $\mathcal{V}_s$. Assuming $\mathcal{V}_s$ is of type $A$, we gather a set of latent vectors: $\{\mathbf{h}_s^{M_1}, \mathbf{h}_s^{M_2}, \ldots, \mathbf{h}_s^{M_K}\}$, where $K$ denotes the number of metapaths associated with $\mathcal{V}_s$. A straightforward approach would be using the average vector. However, to capture the distinct contributions of different metapaths, we employ a similar mechanism to the previous aggregation, enhancing the representational power of the aggregation:

$$\mathbf{h}_s' = \sum_{k=1}^{K} \beta_s^{M_k} \cdot \mathbf{h}_s^{M_k}, \tag{9}$$

where importance weights $\beta_s^{M_k}$ are used to compute a weighted sum of the metapath representations, thereby finalizing the aggregated output for node $\mathcal{V}_s$. This aggregation synthesizes information across different metapaths, allowing the model to highlight more significant metapaths based on their relevance to the target node's type and context. The resulting vector $\mathbf{h}_s'$ represents a comprehensive embedding of $\mathcal{V}_s$, considering both its type-specific properties and connectivity in the graph.

After two-level aggregation, each token becomes an updated representation $\{\mathbf{h}_i', \forall i \in \{\mathcal{V}\}\}$.

## 3.4 Inner Ordering & Updating

After aggregating each token into an updated representation, we start to capture long-range heterogeneity and homogeneity dependencies among the graph. Firstly, all nodes are grouped by their types and ordered within each group based on the number of metapath instances. This ordering reflects an assumption that the greater the number of metapath instances associated with a node, the more significant that node is within its type. After that, each Mamba block updates node representations within each node type following the ordered sequence.

### 3.4.1 Inner-Type Ordering

**The Need for Ordering in Graphs.** Graphs are unordered structures, where the arrangement of nodes does not imply any specific processing sequence. However, for SSM-based models such as Mamba, which are adapted from architectures designed for sequential data, converting the graph data into an ordered sequence is imperative. Ordering nodes provide a structured way to input data into these models, enabling the application of advanced architectures developed for ordered data.

**Efficient Ordering Method.** Given the unordered graph data, proposing an efficient method to order nodes is essential. This strategy addresses it by grouping nodes according to type and then ranking them within their groups based on their number of metapath instances:

$$\mathcal{V}_A = \{\mathcal{V}_i \in \mathcal{V} : \text{type}(\mathcal{V}_i) = A\}, M_i = \sum_{M \in \mathcal{M}} \text{count}(\mathcal{V}_i, M). \tag{10}$$

Nodes $\mathcal{V}_i$ in each group $\mathcal{V}_A$ are ordered in ascending order of $M_i$. Because each node in the Mamba block can only gather information from previous nodes, the most influential nodes are placed towards

the end of the sequence, ensuring that they have the maximal possible context for their feature representations, enhancing the model's effectiveness. This metric not only reflects the quantitative involvement of nodes in graph structures but also prioritizes nodes with higher connectivity and semantic importance within the graph. This prioritization is aligned with the model's need to capture the most relevant information early in its operations.

By ordering the nodes in this manner, we harness the complex and rich information present in heterogeneous graphs and adapt it for models requiring sequential data input, thereby bridging the gap between unordered graph data and the ordered input requirements of SSMs.

### 3.4.2 INNER-TYPE UPDATING

**Context-Aware Filtering in Mamba.** Mamba optimizes the processing of graph data by integrating a selective filtering mechanism that distinguishes between relevant and irrelevant contexts. This functionality is critical for managing the extensive and varied connections typical in graphs, enabling the model to maintain focus on pertinent information over extended sequences.

**Mechanism for Dynamic Information Filtering.** Mamba employs the matrices $\overline{\mathbf{A}}$ and $\overline{\mathbf{B}}$ which are discretized coefficients in Equation (2) to modulate the influence of past states and current inputs: $\mathbf{h}_t = \overline{\mathbf{A}}h_{t-1} + \overline{\mathbf{B}}x_t$ where $x_t$ is an input sequence. These matrices enable the model to 'remember' or 'forget' information, facilitating the management of long-range dependencies where the relevance of information varies across the graph.

**Sparsification of Graph Attention.** The model enhances traditional graph attention mechanisms by sparsifying attention, focusing computational resources on the graph's most informative parts to reduce complexity and enhance performance:

$$\mathbf{y} = SSM(\overline{\mathbf{A}}, \overline{\mathbf{B}}, \mathbf{C})(x) \tag{11}$$

where $SSM(\cdot)$ refers to output formed by $y_t = \mathbf{C}h_t$ in Equation (3). $\mathbf{C}$ is a parametrized projection of the input which decides whether the state is included in the output $y_t$. $SSM(\cdot)$ aggregates significant features across the graph, implementing context-aware filtering. While it cannot utilize the convolutional kernel in Equation (4) anymore due to selectivity, it leverages hardware-aware state expansion to overcome efficiency limitations(Gu & Dao, 2023).

**Implications.** (1) *Enhances efficiency*: Reduces computational overhead by focusing on critical interactions; (2) *Improves accuracy*: Decreases noise in data, increasing prediction accuracy; (3) *Preserves long-range dependencies*: Ensures critical nodes are maintained over large graph.

Overall, Mamba's updating for context-aware filtering enables better management of graph data by focusing on relevant features, which is crucial for handling complex network structures.

### 3.5 OUTER ORDERING & UPDATING

After utilizing $|\mathcal{A}|$ Mamba to update nodes from $|\mathcal{A}|$ distinct node types in the previous step, we capture the long-range homogeneity dependencies within each node type. Subsequently, our focus shifts to addressing the long-range heterogeneity dependencies that exist across different node types. We design a new Ordering & Updating strategy for this purpose, as detailed below.

### 3.5.1 CROSS-TYPE ORDERING

**Extending Ordering Across Node Types.** Having established an order within individual node types, we now extend it to span across different node types. This outer ordering is crucial for capturing heterogeneous long-range dependencies reflecting global structural patterns within the graph.

**Cross-Type Ordering Strategy Based on Degree.** To achieve this ordering, we prioritize nodes based on node degree. An overarching sequence is then determined, which indicates their global influence and connectivity. This sequence ensures that nodes acting as critical hubs between different node types are positioned to maximize their contextual influence throughout the graph.

**Implementation of Heterogeneous Dependencies.** By arranging nodes to maximize within-type relationships and then leverage between-type interactions, we capture the full spectrum of dependen-

cies within the graph. It enhances the model's ability to process heterogeneous data and aligns with Mamba's capability to handle complex inter-dependencies inherent in large-scale graph structures.

### 3.5.2 CROSS-TYPE UPDATING

**Global Context-Aware Updating.** With the outer ordering, updating node representations across different types involves an approach to integrate and filter information from global sources. This step is essential for managing the diversity of interactions among heterogeneous node types.

**Dynamic Cross-Type Information Filtering.** Utilizing structured matrices $\overline{\mathbf{A}}$ and $\overline{\mathbf{B}}$, the model now updates each node's state considering its degree and type-specific context: $\mathbf{h}_t^{global} = \overline{\mathbf{A}} h_{t-1}^{global} + \overline{\mathbf{B}} x_t^{cross}$, where $x_t^{cross}$ represents the aggregated input features from multiple node types, emphasizing the integration of diverse information streams.

**Enhanced Graph Attention for Heterogeneity.** Attention mechanisms are adapted to dynamically allocate focus across different node types, balancing intra-type precision with inter-type coverage: $y_t = \mathbf{C} h_t^{global}, \mathbf{y} = SSM(\overline{\mathbf{A}}, \overline{\mathbf{B}}, \mathbf{C})^{global}(x^{cross})$, where $SSM(\cdot)^{global}$ extends the $SSM(\cdot)$ to encapsulate interactions across node types, refining the model's ability to synthesize and highlight key features from a graph-wide perspective.

**Outcomes.** Updated nodes' representations by combining information from all node types.

## 4 EXPERIMENTS

In this section, we compare our model with many state-of-the-art baselines on standard heterogeneous graph benchmark (HGB) (Lv et al., 2021b) and long-range heterogeneous graph benchmark (LRHGB) (Hu et al., 2021).

### 4.1 DATASETS AND BASELINES

**Long Range Heterogeneous Graph Benchmark.** We select the large-scale dataset `ogbn-mag` from OGB challenge (Hu et al., 2021). `ogbn-mag` is a heterogeneous graph derived from a subset of the Microsoft Academic Graph (MAG). It comprises four kinds of entities — papers, authors, institutions, and fields of study — along with four types of directed relationships linking two different entity types.

**Standard Heterogeneous Graph Benchmark.** We choose three heterogeneous graphs including `DBLP`, `IMDB`, and `ACM` from HGB benchmark (Lv et al., 2021b) on the node classification task. `DBLP` is an online bibliography resource for computer science. We employ a frequently utilized subset spanning four domains, where nodes represent authors, papers, terms, and venues. `IMDB` is a portal dedicated to movies and associated details. We utilize a subset covering genres such as Action, Comedy, and Drama. `ACM` is a citation network as well. We utilize the subset provided in HAN (Wang et al., 2021), maintaining all connections including paper citations and references.

We choose MLP(Hu et al., 2020a), GraphSAGE(Hamilton et al., 2017), RGCN(Schlichtkrull et al., 2018), NARS(Yu et al., 2020), HGT(Hu et al., 2020b), SeHGNN(Yang et al., 2023), LMSPS(Li et al., 2024a), RSHN(Zhu et al., 2019), HetSANN(Hong et al., 2019), GTN(Yun et al., 2020), HGT(Hu et al., 2020b), Simple-HGN(Lv et al., 2021a), HINormer(Mao et al., 2023), RGCN(Schlichtkrull et al., 2018), HetGNN(Zhang et al., 2019), HAN(Wang et al., 2021), MAGNN(Fu et al., 2020), SeHGNN(Yang et al., 2023), DiffMG(Ding et al., 2021), PMMM(Li et al., 2022), and Graph-Mamba (Wang et al., 2024) (adding node-type linear projection at the head) as our baselines.

### 4.2 BENCHMARK EVALUATION

**Comparison on Performance.** We evaluate the performance of our **HGMN** against 19 heterogeneous graph baselines using accuracy and F1 score metrics across several datasets shown in Table 1. Our model demonstrated superior performance, achieving the highest scores on the ogbn-mag, DBLP, and ACM datasets with an accuracy of $0.5763 \pm 0.0043$ on ogbn-mag, and F1 scores of $0.9602 \pm 0.0010$

Table 1: **Performance of Node Classification.** We conduct an experiment using 19 baselines of heterogeneous graph modeling methods, evaluated with the F1 score and accuracy. Across all configurations, HeteGraph-Mamba achieves the best performance in the obgn-mag, DBLP, IMDB and ACM datasets. Highlighted are the top first, second, third.

| Model | ogbn-mag | DBLP | IMDB | ACM |
|---|---|---|---|---|
| | Accuracy | F1 score | F1 score | F1 score |
| MLP | 0.2692±0.0026 | 0.4933±0.0030 | 0.2815±0.0033 | 0.4018±0.0104 |
| GCN | 0.3489±0.0019 | 0.9084±0.0032 | 0.5273±0.0024 | 0.8743±0.0135 |
| GAT | 0.3767±0.0025 | 0.9142±0.0027 | 0.5380±0.0094 | 0.8845±0.0136 |
| GraphSAGE | 0.4678±0.0067 | 0.9182±0.0011 | 0.5591±0.0036 | 0.8983±0.0098 |
| RGCN | 0.4737±0.0048 | 0.9207±0.0050 | 0.6205±0.0015 | 0.9141±0.0075 |
| RSHN | 0.4728±0.0031 | 0.9381±0.0055 | 0.6422±0.0103 | 0.9032±0.0154 |
| HetSANN | 0.4781±0.0029 | 0.8056±0.0150 | 0.5768±0.0044 | 0.8991±0.0037 |
| NARS | 0.5088±0.0012 | 0.8845±0.0034 | 0.6539±0.0018 | 0.9310±0.0040 |
| MAGNN | 0.4926±0.0024 | 0.9376±0.0045 | 0.6467±0.0167 | 0.9077±0.0065 |
| HetGNN | 0.4732±0.0130 | 0.9176±0.0043 | 0.4825±0.0067 | 0.8591±0.0025 |
| HAN | 0.5037±0.0066 | 0.9205±0.0062 | 0.6463±0.0058 | 0.9079±0.0043 |
| DiffMG | 0.5157±0.0044 | 0.9420±0.0036 | 0.5975±0.0123 | 0.8807±0.0304 |
| PMMM | 0.5188±0.0031 | 0.9514±0.0022 | 0.6758±0.0022 | 0.9371±0.0017 |
| Simple-HGN | 0.5215±0.0013 | 0.9446±0.0022 | 0.6736±0.0057 | 0.9335±0.0045 |
| SeHGNN | 0.5671±0.0014 | 0.9524±0.0013 | 0.6821±0.0032 | 0.9367±0.0050 |
| GTN | 0.5392±0.0037 | 0.9397±0.0054 | 0.6514±0.0045 | 0.9120±0.0071 |
| HGT | 0.4929±0.0061 | 0.9349±0.0025 | 0.6720±0.0057 | 0.9100±0.0076 |
| HINormer | 0.5520±0.0062 | 0.9494±0.0021 | 0.6783±0.0034 | 0.9379±0.0122 |
| Graph-Mamba++ | 0.5026±0.0020 | 0.9271±0.0019 | 0.6322±0.0096 | 0.9017±0.0108 |
| **HGMN** | **0.5763±0.0043** | **0.9602±0.0010** | **0.6917±0.0028** | **0.9484±0.0046** |
| - w/o Inner-ordering | 0.5371±0.0034 | 0.9410±0.0041 | 0.6535±0.0079 | 0.9112±0.0130 |
| - w/o Outer-ordering | 0.5237±0.0040 | 0.9342±0.0031 | 0.6492±0.0036 | 0.9209±0.0085 |

on DBLP, and $0.9484 \pm 0.0046$ on ACM. It also performed strongly on IMDB with an F1 score of $0.6917 \pm 0.0028$. These results highlight **HGMN**'s robustness and its ability to manage complex graph structures, validating its advanced node classification capabilities in heterogeneous graphs.

**Comparison on Efficiency.** In our study, we evaluate the time and memory requirements of various HGNNs on the DBLP dataset shown in Figure 2. This analysis highlights our model's efficiency, achieving an excellent F1 score with moderate memory use and computational time. Other models like SeHGNN and HINormer, although high-performing, required significantly more memory. This evaluation emphasizes our model's capability to efficiently process large-scale graphs without compromising performance.

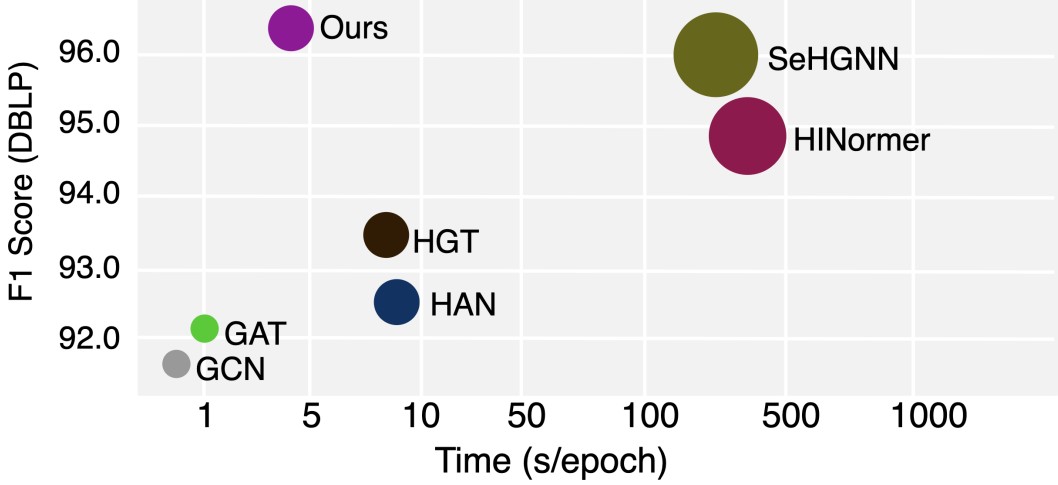

Figure 2: **Time and Memory.** The area of the circles represents the (relative) memory consumption.

## 5   CONCLUSION

In this study, we present **HGMN**, a cutting-edge network that utilizes SSSMs to redefine heterogeneous graph learning, outperforming transformer-based approaches. Our strategy transfers unordered graph data into a two-level sequential format. The first level organizes graph data to capture homogeneity dependencies among similar node types and subsequently leverages organized sequences to address heterogeneity dependencies across different types of nodes. This dual sequential process not only enhances the expressiveness and computational efficiency of the linear-time performance, but also improves the accuracy of node embeddings for downstream tasks. Our experimental results validate HGMN's superior capability in managing the complexities of large heterogeneous graphs, demonstrating marked performance improvements over existing methods on benchmarks. HGMN also has limitations on theoretical analysis, which can provide more strong proofs.

### ETHICS STATEMENT

Our research methodology can bolster comprehension and problem resolution across numerous areas, including AI research, fostering clearer and more decipherable outcomes. However, this method might simplify intricate problems too much by dividing them into distinct segments, possibly neglecting subtleties and linked components. Moreover, a strong dependence on this approach could curtail innovative problem-solving, since it promotes a sequential and orderly method, which may hinder unconventional thought processes.

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

# A  APPENDIX

## A.1  SYSTEM

All experiments are carried out on a High Performance Computing cluster. There are 34 GPU nodes where 16 nodes each have 2 NVIDIA 40GB Tesla A100 PCIe GPUs, 52 CPU cores, and 192 GB of CPU RAM while 18 nodes are each equipped with 4 NVIDIA 80GB Tesla A100 SXM GPUs, 52 CPU cores, and 512 GB of CPU RAM. The driver version 525.105.17 on these nodes is compatible with CUDA 12.0 or earlier. The operating system is Red Hat Enterprise Linux 7.9.

## A.2  HYPERPARAMETER

In our experiments, we require hyperparameters for tokenization, heterogeneity alignment (Num heads and Metapath Attention Dim are exclusive to this step), and the MAMBA layers. We use the Adam optimizer with weight decay. Below, we provide a list of all the hyperparameters used in our experiments.

Table 2: Hyperparameter used in the task.

| Hyperparameter | ogbn-mag | DBLP | IMDB | ACM |
|---|---|---|---|---|
| Num Layers | 4 | 2 | 2 | 2 |
| Hidden Dim | 128 | 64 | 64 | 64 |
| Learning Rate | 0.003 | 0.0005 | 0.0005 | 0.0005 |
| Weight Decay | 0.0005 | 0.0001 | 0.0001 | 0.0001 |
| Num Epochs | 300 | 150 | 150 | 150 |
| Num heads | 8 | 8 | 8 | 8 |
| Metapath Attention Dim | 256 | 128 | 128 | 128 |

