# OpenReview forum: "HeteGraph-Mamba: Heterogeneous Graph Learning via Selective State Space Model"
_ICLR.cc/2025/Conference — ICLR 2025 Conference Withdrawn Submission_

### Official Review · Reviewer_Byh9 · 2024-10-21

**Soundness:** 2
**Presentation:** 2
**Contribution:** 2
**Rating:** 5
**Confidence:** 4

**Summary:**

This paper introduces HGMN, the first to apply selective state space models (SSSMs) to heterogeneous graph learning. HGMN captures long-range dependencies and adapts SSSMs for heterogeneous data. It outperforms 19 state-of-the-art methods in both accuracy and efficiency.

**Strengths:**

The paper's description of the model is relatively clear. The language is fluent.

**Weaknesses:**

1. The problem statement is unclear. The introduction does not effectively clarify whether the focus is on learning heterogeneous graph representations or capturing long-range dependencies. I suggest the authors clearly delineate the specific focus and differentiate between these two aspects in the introduction.

2. Lack of sufficient motivation. The paper introduces challenges related to selective state space models (SSSMs) but fails to explain the necessity of integrating SSSMs. I recommend the authors elaborate on the limitations of existing approaches and provide examples where SSSMs would be particularly beneficial to enhance the motivation.

3. The experimental results are not convincing:

(1) The paper mentions two recent baselines—LMSPS (Li et al., 2024a) and Graph-Mamba (Wang et al., 2024)—but LMSPS is not included in the experiments. The authors should explain why LMSPS was omitted and discuss its relevance for comparison.

(2) Graph-Mamba focuses on long-range graph sequence modeling rather than heterogeneous graph representation learning. The authors should justify the appropriateness of using it as a baseline.

(3) The paper claims that HGMN performs well on large heterogeneous graphs, yet the four datasets used are all standard datasets, not large-scale ones. I suggest the authors consider using specific large-scale heterogeneous graph datasets to better support their claims regarding HGMN's performance.

**Questions:**

Please see Weaknesses.

---

### Official Review · Reviewer_T29q · 2024-10-31

**Soundness:** 2
**Presentation:** 3
**Contribution:** 2
**Rating:** 3
**Confidence:** 5

**Summary:**

The paper presents HeteGraph-Mamba (HGMN), a novel heterogeneous graph learning framework that uses Selective State Space Models (SSSMs) to address the limitations of existing graph neural networks. HGMN introduces a six-step process, including graph tokenization and context-aware node ordering, to improve computational efficiency and capture long-range dependencies across heterogeneous nodes. Experimental results indicate that HGMN achieves higher accuracy and efficiency than other state-of-the-art models on several benchmark datasets.

**Strengths:**

S1：The integration of SSSMs in heterogeneous graph learning is innovative, addressing scalability and long-range dependency challenges not fully tackled by traditional Transformer-based models. HGMN’s tokenization and ordering mechanisms for heterogeneous nodes offer a fresh approach to sequence modeling in graphs.
S2：The six-step processing flow in HGMN separately addresses homogeneity and heterogeneity dependencies, enhancing the model's expressive capability. Additionally, the intra-type and inter-type node ordering improves the accuracy of context capture.
S3：The paper provides a detailed overview of each component of the HGMN framework, making it easy to follow the flow and understand the contributions of each design choice.
S4：HGMN’s performance improvements over 19 baseline models on accuracy and efficiency metrics demonstrate the potential impact of this approach in the heterogeneous graph learning domain.

**Weaknesses:**

W1. The motivation for using Mamba on graphs is unclear. Mamba is faster than Transformer only when modeling sequences longer than 2k.
W2. Although the paper demonstrates the superiority of the experimental results, it lacks sufficient theoretical analysis of the proposed method, especially in the theoretical basis for improving the efficiency and accuracy of SSSMs. It is recommended to add more mathematical analysis or complexity proof.
W3. The previous experiments only verified the effect of HGMN in the node classification task. The applicability and performance of other graph tasks (such as edge prediction and graph classification) have not been tested, which affects its wide applicability.
W4. The model hyperparameters are not explained enough, and the analysis of the functions of each module is also lacking. It is recommended to increase the analysis of hyperparameters and add ablation experiments.

**Questions:**

Q1: Why was Mamba chosen for graphs? Mamba shows improved efficiency over Transformers when modeling sequences longer than 2K tokens, based on current knowledge. Providing a theoretical analysis to support this choice in graph contexts would add significant value to the paper.
Q2: Are all six processing steps necessary? A more detailed ablation study on each step would better clarify the contributions and necessity of each component in the model.
Q3: Performance on tasks beyond node classification: Can the framework achieve similar high performance on other tasks beyond node classification?

---

### Official Review · Reviewer_bL8y · 2024-11-03

**Soundness:** 2
**Presentation:** 2
**Contribution:** 2
**Rating:** 3
**Confidence:** 4

**Summary:**

This paper applies Mamba to heterogeneous graph learning tasks. The authors provide a full recipe to build the model, which includes tokenization, heterogeneity alignment, inner and outer ordering, and inner and outer updating steps. The six-step recipe help convert the heterogeneous graph to sequence, capture long-range dependency among various types of nodes as well as reduce computation burden. The empirical results demonstrate that the proposed HGMN enjoys higher accuracy and more efficiency than exsiting transformer-based GNNs on multiple heterogeneous graph benchmarks

**Strengths:**

1. The proposed recipe to implement the HGMN is highy scalable and take both the long-range homogeneity and heterogeneity into take into consideration.

**Weaknesses:**

1. The motivation is too weak. Since Mamba claims to handle long-range context better
and operates in linear time complexity, the paper applies Mamba to heterogeneous graph
learning tasks, expecting superior results. This point lacks persuasiveness.
2. Novelty is very limited. The paper shares many similarities with  Graph Mamba（KDD 2024）.

2.1 For instance, in the Tokenization section, both fundamentally encode subgraphs to
represent nodes, with the difference being that Graph Mamba uses a random walk
approach to sample subgraphs, while this paper uses metapath instances.

2.2Additionally, the language style in the Tokenization section of the article is very similar
to that of Graph Mamba.

2.3 In this paper, I think the only difference between it and Graph Mamba lies in the inner
and outer ordering methods. However, these ordering methods lack persuasiveness.
*Why does the inner ordering rely on the number of metapaths as its basis?
*Why can’t the number of node degree be used as a basis during inner ordering?
*Shouldn’t the importance of different metapaths be considered?

3. Some important detail  is missing. For example, in the Metapath Instance Aggregation
section, the implementation of the instance encoder is not explained. How is beta
implemented in Equation 9?

4. The experiments are not enough. The ablation experiments are not
analyzed and are not comprehensive, for example, there is a lack of ablation on the
TOKENIZATION and HETEROGENEITY ALIGNMENT sections. There is also a lack of
analysis on key parameters in the method.


Reference:*Graph Mamba: Behrouz, Ali, and Farnoosh Hashemi. "Graph mamba: Towards learning on
graphs with state space models." Proceedings of the 30th ACM SIGKDD Conference on
Knowledge Discovery and Data Mining. 2024.

**Questions:**

1. How can you prove that your tokenization is both effective and efficient？
2. For Equation 6, is it reliable to map vectors of different types of nodes to the same latent
space using just a single linear layer? If this approach is successful, does it suggest that
the node feature vectors xi constructed in this paper already exist in a larger latent space,
and that a simple linear layer is used for dimension reduction? Could this operation
interfere with subsequently capturing homogenous and heterogeneous dependencies
within the scope?
3. In the ordering process, if multiple nodes have the same degree or the same number of
relevant metapaths, how should they be handled?

---

### Official Review · Reviewer_oSMa · 2024-11-04

**Soundness:** 2
**Presentation:** 2
**Contribution:** 2
**Rating:** 3
**Confidence:** 3

**Summary:**

This paper proposes a graph mamba network as a graph model to learn node representations, aiming to incorporate long-range dependencies for heterogeneous graphs. It adopts subgraphs containing multiple meta-path instances to be tokens of all nodes. Aggregate embeddings of meta-path instances for node $v$ within one meta-path and across multiple meta paths using attention mechanism to get the node embedding and then uses the updated method in MAMBA to update node embeddings within one node type first and then across node types to model the long-range dependency. The experimental results show marginal improvement.

**Strengths:**

1. it is good to adopt MAMBA for graph data.
2. Comprehensive baseline models are used.
3. The proposed model is relatively fast empirically.

**Weaknesses:**

1. The related work lacks targeting the problem that the paper wants to solve. i.e., if you claim the proposed model can do long-range dependency, so your related work should discuss the work that is related to this challenge.
2. The paper has typos and doesn't provide explanations for some symbols, i.e., $\mathcal{N}_s^M$, and `spacial` at line 172.
3. Additionally, the experiments are unable to be reproducible. it is unclear how metapaths are built. Since the improvement is marginal, it would be better to show the variance of the performance. This paper lacks an ablation study for the components in the model. sin
4. My major concern is that the local information of a node might gradually disappear during the inner and outer updates since the order of these two processes is only determined by the degree.

**Questions:**

1. How do you get the subgraph for all nodes? Does it include every node next to it?
2. Also please address the fourth point in weakness.

---

### Note · Authors · 2024-11-20

**Comment:**

Thanks so much for ever reviewer's time and important comments! We think our current submission still needs a major revision, so, we decide to withdrawl!

Authors.

**Withdrawal Confirmation:**

I have read and agree with the venue's withdrawal policy on behalf of myself and my co-authors.